# The Effect of Hydrothermal, Microwave, and Mechanochemical Treatments of Tin Phosphate on Sorption of Some Cations

**DOI:** 10.3390/ma16134788

**Published:** 2023-07-03

**Authors:** Oleg Zakutevskyy, Svitlana Khalameida, Volodymyr Sydorchuk, Mariia Kovtun

**Affiliations:** Institute for Sorption and Problems of Endoecology NAS of Ukraine, General Naumov Str. 13, 03164 Kyiv, Ukraine; zakutevskyy@i.ua (O.Z.); bilychi@ukr.net (V.S.); mariakovtun96@gmail.com (M.K.)

**Keywords:** tin phosphate, modification, adsorption, Cs, Sr, U

## Abstract

The samples of precipitated tin (IV) phosphate, modified using hydrothermal, microwave, and mechanochemical procedures, were studied in the process of Cs(I), Sr(II), and U(VI) ion sorption. The initial and modified samples were investigated before and after sorption using XRD, XRF, FTIR, and nitrogen adsorption–desorption. It was found that the modification procedures transformed the micro-mesoporous structure of the initial sample into a meso–macroporous structure. As a result, the sorption capacity in relation to all ions increases several times. This indicates the determining role of the porous structure, primary content, and mesopore size on the sorption activity of tin phosphate. The samples, treated in the form of a wet gel, which is a novel procedure, showed the maximum sorption indicators. The sorption of all the tested ions is described by the Langmuir isotherm.

## 1. Introduction

Sorption technologies are traditionally used to treat polluted water sources. Their efficiency depends on the physicochemical characteristics of the used sorbents [1,2,3,4,5]. Furthermore, these processes are of great scientific and technological interest, mainly due to the high degree of controllability, flexibility, and reversibility of the sorption phenomena [4]. The increased attention paid to inorganic ion exchangers is associated with the instability of organic resins to high temperatures and strong ionizing ability [1]. Therefore, inorganic materials are promising for the removal of radionuclides from wastewater [5,6,7,8,9]. Due to Bronsted acidity, the amorphous and crystalline tetravalent metal phosphates as water-insoluble salts have many characteristics that are important from a practical point of view, namely, thermostability, resistance to acid, alkali and oxidation media, as well as high ion exchange capacity and sorption affinity, including to Cs(I), U(VI), and Sr(II) ions [1,2,3,7,10].

The tin (IV) phosphate SnP belongs to the family of phosphates that are typical cation exchangers [1,2,7,8,9,10,11,12,13]. Indeed, tin phosphate with various P/Sn ratios has attracted attention for a long time as a material for the sorption and separation of cations [7,8,9,11], including for cation removal from nuclear waste solutions [1,5,12,13,14]. Recent studies have confirmed that the SnP of composition Sn(HPO_4_)_2_ nH_2_O is a promising potential ion exchanger [5,15,16,17,18,19,20]. However, unlike zirconium and titanium phosphates, there are only separate studies devoted to the adsorption of cesium and strontium cations using tin phosphate [3,5,13] or its composition with zirconium phosphate [21]. In particular, the sorption of Cs(I) and Sr(II) on the mixed tin–zirconium phosphate prepared via the sol–gel method was studied using the batch technique [21]. However, the authors of these works pay attention to the conditions of the sorption of these cations but do not sufficiently study the influence of the physicochemical characteristics of adsorbents, the porous structure in particular, on their sorption properties. The study of this aspect was attempted by the authors of References [22,23,24,25,26,27] using tin, titanium, and zirconium dioxides and phosphates for the sorption of some cations.

At the same time, it is well known that the surface, porous and crystal structure, including the nature and content of active sites, are decisive parameters ensuring effective adsorbent application [1,2,3,14,26]. Among other things, the specific surface area, as well as the volume and size of pores and their distribution of sizes, are the most important characteristics on which the access of the hydrated ions to the ion-exchanged centers of the surface, and consequently, the sorption capacity and kinetics of this process, depend. The latter is especially crucial for precipitated tin phosphate with a high micropore content [9,14,20,28]. Obviously, mesoporous or meso–macroporous adsorbents have optimal properties facilitating the elimination of diffusion limitations [22,29,30,31,32,33,34]. As shown in studies [31,32,33,35,36,37], hydrothermal, microwave, and mechanochemical treatments provide the formation of just such a porous structure and, as a consequence, an improvement in the adsorption properties of some materials. The results of References [22,23] are remarkable in that the microwave and mechanochemical treatments of tin dioxide contribute to a significant increase in the sorption capacity, as well as the distribution coefficient of cesium, strontium, and uranium cations. The main reason for this is an increase in the volume and size of the mesopores, even with a reduction in the specific surface area measured using nitrogen adsorption. At the same time, surface sites become more accessible for hydrated cations due to the development of the mesoporous structure.

Moreover, the use of these procedures of modification makes it possible to adjust the characteristics of the porous structure for tin phosphate within wide limits [28]. A new approach based on the modification of the wet gel before its drying showed particular effectiveness. However, the porous structure of ion exchangers based on SnP and their influence on the sorption of cations from aqueous solutions have not been previously studied. Only some regularities have been established that characterize the effect of the parameters of the porous structure on the catalytic and electrochemical properties of tin phosphate [28,34,35,36,37,38,39]. These results confirm the assumption about the decisive importance of the mesoporous fraction for its effective application.

Therefore, the aim of this work is to study the porous structure’s influence on the adsorption of cesium, strontium, and uranium cations on tin phosphate, modified using different procedures. It is obvious that studies of the sorption of these cations are of particular interest from a scientific and practical point of view since their radioactive isotopes are components of the “block” water of the Chernobyl and Fukushima nuclear power plants [6,40,41]. This prevents their processing using technologies for the purification of water contaminated with radioactive waste.

## 2. Materials and Methods

### 2.1. Adsorbents

The initial sample SnP was prepared via precipitation, as described elsewhere [18,28]. This procedure was as follows. Tin phosphate was synthesized at pH 1 in the form of a gel-like precipitate by means of the gradual addition of a 6% aqueous H_3_PO_4_ solution to a 0.3 M aqueous solution of SnCl_4_·5H_2_O. The ratio of P/Sn in the solution was 2. After aging in the mother liquor for 20 h, the gel was washed with distilled water up to an absence of impure ions and filtered to a moisture content of 90%. Part of the gel was dried in air at 20 °C for 50 h for the preparation of the xerogel. Subsequently, the SnP was modified using different treatments, such as wet gel and dried xerogel.

Hydrothermal treatment (HTT) was performed in the form of wet gel and dried xerogel in the steel autoclaves with a volume of 45 mL at 150–300 °C and autogenous pressure for 3 h. A “NANO 2000” high–pressure reactor (Plazmatronika, Wroclaw, Poland) with the power of 650 W was used for the microwave treatment (MWT). The duration, temperature, and pressure of MWT were 0.5 h, 185 °C, and 5 MPa (MWT of wet gel), as well as 0.5 h, 250 °C, and 9 MPa (MWT of dried xerogel). The mechanochemical treatment (MChT), in the form of wet gel and dried xerogel with the addition of water, was carried out at the rotation speed of 600 rpm for 0.5 h using a planetary ball mill Pulverisette-7, premium line (Fritsch Gmbh, German), with a vessel of silicon nitride. A total of 25 balls from S_3_N_4_ with a 10 mm diameter (total ball mass–40 g) were used as working bodies. All treated samples were dried at 20 °C for 72 h.

### 2.2. Physicochemical Methods of Investigations

The porous structure of SnP samples was studied using low-temperature nitrogen adsorption–desorption. The isotherms were obtained using an automatic gas adsorption analyzer ASAP 2405 N (“Micromeritics Instrument Corp.”, Norcross, GA, USA) after outgassing the samples at 150 °C for 2 h. The specific surface area S, volume of mesopores V_me_, and volume of micropores V_mi_ were calculated from these isotherms using the BET, BJH, and t-methods, respectively. The total pore volume V_Σ_ was determined by means of impregnation of granule samples, dried at 150 °C with liquid water. The volume of macropores V_ma_ is the difference between V_Σ_ and (V_me_ + V_mi_). The curves of the pore size distribution (PSD) were plotted using the desorption branches of isotherms.

The atomic content of Sn and P in the initial and modified sample calculation was based on X-ray fluorescent analysis XRF (Axios mAX PANalytical, Wyatt Technology, Santa Barbara, CA, USA). Some tested samples before and after sorption of cations were studied by means of X-ray powder diffraction (XRD) using Philips PW 1830 diffractometer with CuK_α_-radiation. FTIR spectra for the same samples were recorded in the range 400–4000 cm^−1^ using “Spectrum-One” spectrometer (Perkin-Elmer, Waltham, MA, USA). Mixture of samples with KBr, calcined at 600 °C, in the ratio of 1:20 was used for these measurements. 

### 2.3. Testing the Sorption Properties

As mentioned above, studying the effect of various types of processing on the porous structure and, as a result, on the sorption characteristics of tin phosphate is the main goal of this study. Based on the results of Reference [18], samples for sorption measurements were selected. They simultaneously have a meso- or meso–macroporous structure and a sufficiently high specific surface area, but a minimum content of micropores (Table 1). The selection was made for each type of modification (HTT, MWT, and MChT) and its variant (treatment in the form of wet gel and dried xerogel). As can be seen, mesopores and macropores volume for some modified samples reaches 0.05–0.38 and 0.08–0.38 cm^3^/g. In addition, modification procedures used allow the preparation of samples with larger diameters of mesopores compared with initial sample.

The sorption properties of the tin phosphate were characterized by sorption capacity *A* and distribution coefficient *K_d_*. The studies were performed under static conditions; ratio of solid/liquid was 1:2000. The pH value was adjusted to 5.2 ± 0.2 (without background) during sorption of ions U (VI), Cs (I), and Sr (II) [10,22]. Under these conditions, studied metals are present in the solution, mainly in the form of cations. The solutions with an initial concentration (C_0_) of 0.2 meq/L for Cs^+^ and Sr^2+^ ions and 1.0 meq/L for UO_2_^2+^ were used. The changes in concentrations of Cs and Sr in the solution were determined using an atomic absorption spectrophotometer AA-6300 (Shimadzu, Kyoto, Japan). The uranium concentration in solution was determined on a KFK-3 photometer by the technique based on formation by the UO_2_^2+^ ion of a stable colored complex with arsenazo III at pH 1 [42].

## 3. Results and Discussion

### 3.1. Composition and Crystal Structure of Adsorbents

According to the data of the XRF analysis, the P/Sn ratio is 1.98 for the initial sample. Modification practically does not change this ratio: it is within 1.93–1.97 for modified samples, which corresponds to tin hydrophosphate Sn(HPO_4_)_2_. This is previously confirmed by the results of DTA-TG and XRD data [28]. The initial sample is X-ray amorphous (Figure 1). In addition, all the samples subjected to MWT as well as HTT at 300 °C have the low-crystalline structure of Sn(HPO_4_)_2_ (Figure 1). Therefore, the modification procedures used do not change the phase composition: all tested samples have the structure of tin hydrophosphate (in an X-ray amorphous, poorly crystalline state).

It was established that modification of precipitated tin phosphate does not change the content of functional groups [28]. For example, the FTIR spectra for samples modified by MChT of the wet gel at 600 rpm (No. 2 from Table 1), which were recorded in the range 2000–4000 cm^−1^, are presented in Figure 2. As known, the absorption bands (a.b.) that appear in this spectral range are related to the vibration of surface groups. Thus, a band at 2400 cm^−1^ is present in these spectra. It is attributed to the (P)–OH stretching vibration [37,39,43,44,45,46,47].

These groups are cation exchange sites. The intensity of the indicated band is practically the same for all tested samples, and the intensity of the broad band is in the range of 3300–3500 cm^−1,^ which corresponds to the stretching vibration of surface OH groups [28]. Therefore, the results of XRD, XRF, and FTIR studies show that phase and chemical composition, as well as surface structure, do not undergo significant changes due to the modification procedures used.

### 3.2. Porous Structure of Adsorbents

Unlike the crystal structure, the porous structure of SnP changes significantly as a result of treatments by means of MChT, MWT, and HTT, as was shown in Reference [28]. As a result, it became possible to widely vary the parameters of the porous structure by changing the treatment conditions. The samples possessing different porous structures were selected for sorption measurements. Indeed, the isotherm of nitrogen adsorption–desorption obtained for the initial sample is characteristic of a micro-mesoporous structure with a weakly pronounced hysteresis loop located in the wide range of relative pressure of nitrogen (Figure 3a). As a result, this sample has a relatively low specific surface area and wide pore size distribution (Table 1, Figure 3b). As reported in Reference [28], bottle-shaped and closed pores are present in its structure. The examples of isotherms presented in Figure 3a indicate the development of porosity, firstly, of mesoporous fraction, due to modification. Therefore, the isotherms of IV type for samples modified under hydrothermal conditions (after HTT and MWT), which are characteristic of uniform mesoporous materials with hysteresis loops, were obtained.

Table 1 shows a wide range of all parameters of the porous structure for the tested samples. It should be noted that some samples have a very high specific surface area (300–335 m^2^/g), which is uncharacteristic of tin phosphates previously studied in sorption processes. Comparable or higher specific surface area values were obtained only for ordered mesoporous samples prepared using a more complex template method [34,37,45], but these samples were not studied as cation exchangers.

On the other hand, the accessibility of the surface for hydrated cations can be different since these samples differ in their content and size of mesopores (Table 1). The increase in the diameter of mesopores due to modification is clearly seen from the curves of PSD presented in Figure 3b.

### 3.3. Sorption Properties

Table 2 shows the equilibrium sorption performance with respect to Cs and Sr ions, which is established in a solution with an initial concentration of Cs and Sr ions of 0.2 meq/L. One can see that sorption of U(VI) ions is significantly higher than that of Sr(II) and Cs(I) ions. Therefore, a solution with C_0_(UO_2_^2+^) = 0.2 meqv/L is completely purified by all samples. To obtain equilibrium values of sorption capacities with respect to U(VI) ions, an initial solution with C_0_(UO_2_^2+^) = 1.0 meqv/L was used.

As can be seen from Table 2, used modification procedures have a significant effect on the sorption characteristics of tin phosphate. Thus, the initial sample shows a minimum in sorption capacity with respect to strontium ions and a somewhat greater capacity for uranyl ions. At the same time, cesium ions are not sorbed by this sample. All modified samples have higher values, both *A* and *K_d_*, for the sorption of studied cations. The only exception is the sample subjected to MWT in the form of xerogel (No. 5). This sample does not sorb Cs and shows lower sorption capacity in relation to Sr and U compared with the initial sample. Similarly, two other samples modified as xerogel under hydrothermal conditions (Nos. 6, 7) have a low sorption capacity in relation to all three cations. High values of *A* and *K_d_* were obtained only for xerogel milled in water (No. 3).

On the other hand, modification of wet gel results in the improvement in sorption characteristics in all cases. For example, *A* (Cs) and *A* (Sr) values reach maximal magnitudes, specifically 0.20 mEq/g and 0.30–0.32 mEq/g, after MChT and HTT of the wet gel. It is interesting that *A*(U) increases monotonically with increasing the temperature of HTT in the form of a wet gel in the range of 150–300 °C. The distribution coefficient values also change in a similar way. According to their sorption capacity, the studied ions can be placed in the series of U >> Sr > Cs. This sequence is also characteristic of other phosphate-containing sorbents which were tested under the same conditions [10,24].

In Table 2, the percentage of sorption of each cation is given in brackets. The percentage of sorption, as a separate indicator, makes it possible to estimate the value of the equilibrium sorption capacity of the samples in relation to the maximum possible one at a given concentration of ions in the solution. For samples Nos. 3, 10, and 11, it can be seen that the sorption of U(VI) ions from a solution with C_0_(UO_2_^2+^) = 1.0 meq/L occurs almost completely.

All experimental isotherms were processed in the linear coordinates of Freundlich, Langmuir, and Dubinin–Radushkevich model isotherms [21,24,30]. The Langmuir equation showed the largest value correlation coefficient (R^2^ = 0.98–0.99) for all tested ions, as previously reported for titanium phosphate [24]: C_eq_/A_eq_ = C_eq_/Q_0_ + 1/(b∙Q_0_)
where C_eq_—equilibrium concentration (meq/L), A_eq_—the amount of adsorbate in the adsorbent at equilibrium (meq/g), Q_0_—maximum monolayer coverage capacity (meq/g), and b—the Langmuir isotherm constant (L/meq).

Obtained isotherms are characterized by a plateau that corresponds to an equilibrium state where one ion occupies one active site of a sorbent (Figure 4a). This figure shows the experimental sorption isotherms of Cs, Sr, and U(VI) ions on sample No. 8, which has the highest specific surface area and the most developed porous structure. The calculated dependences K_d_ = f(Ceq) obtained on the basis of the numerical coefficients of the Langmuir model are given in Figure 4b.

Table 3 contains the coefficients of the Langmuir equation as well as the Langmuir model equations, obtained on the basis of experimental data in numerical form, and the equations used to calculate the values of distribution coefficient K_d_.

It can be seen that according to the values of sorption capacities in meq/g and the distribution coefficients, the studied ions form the series U > Sr > Cs, which corresponds to the above assumption about the mechanisms of their sorption on the tested tin phosphate sorbents. Thus, Q_0_(Cs) does not exceed the average sorption values characteristic of ion-exchange processes; Q_0_(Sr) is close to the total ion-exchange capacity of a conventional cation exchanger, and Q_0_(U(VI)) is much higher than this value.

### 3.4. Discussion

Sorption characteristics are determined by the physicochemical properties of sorbents, as well as by the mechanism of sorption. In turn, the physicochemical characteristics depend on the conditions used for the synthesis of sorbents and, next, on their modification. As mentioned above, only the parameters of the porous structure undergo significant changes as a result of modification. Table 1 shows that a feature of the initial sample and samples modified by MWT and HTT of xerogel is relatively low values of the specific surface area and total pore volume. Their other feature is related to the shape of the pores, which determines the accessibility of their surface. As noted in Reference [28], all these samples contain bottle-shaped pores, the entrances to which can be inaccessible to large hydrated cations. If the xerogel or gel is milled in water, its framework is destroyed, and a more open porous structure with a larger specific surface area is formed during the next drying. This structure is more accessible to cations. Indeed, the values of the adsorption capacity and distribution coefficient in relation to all cations increase for these milled samples. Similarly, MWT and HTT of the wet gel before it dries also contribute to the formation of a more accessible porous structure, which promotes the improvement in sorption performance. All this demonstrates the determining influence of the parameters of the porous structure on the sorption process. At the same time, it is difficult to establish clearer correlations due to the simultaneous influence of various parameters of the porous structure (specific surface area, volume of different types of pores, and pore size distribution) on the sorption. Obviously, the most important parameter is not the total specific surface area, but the surface that is accessible to cations. Its value is difficult to estimate, but it definitely depends on the pore size, more precisely, on the cation size/pore size ratio, the reduction of which increases the accessible surface area. These are primarily mesopores, since the tested samples are mainly mesoporous, and mesopores make the maximum contribution to the total specific surface. The most revealing in this regard is the effect of increasing the temperature of HTT of the wet gel on the porous structure and, as a result, on the sorption of uranium (samples Nos. 8–11). These relationships are shown in Figure 5.

It can be seen that the adsorption capacity *A*(U) increases as the size of the mesopores increases, although the total specific surface area decreases with the elevation of HTT temperature. These dependencies indicate that the increase in the sorption capacity is caused by the increase in the surface accessible for the cation UO_2_^2+^. It is noteworthy that the increase in sorption capacity with the increase in temperature of HTT is not observed for the ions Cs(I) and Sr(II) in this row. This is due to the significantly smaller size of these cations. It should be noted that close values of the sorption capacity in relation to Cs(I), Sr(II), and U(VI) were obtained for meso–macroporous titanium phosphate in Reference [24].

The second factor that determines the value of the sorption capacity and the difference between its values for the tested cations is the sorption mechanism. The two mechanisms that are discussed the most often for heavy metal sorption by phosphates are the ion exchange and the dissolution–precipitation processes [48,49,50,51,52]. The possibility of implementing the second mechanism is the ability to form insoluble compounds with phosphate anions during the sorption of cations. Taking into account this condition, the precipitation mechanism is the most likely during uranium sorption. Indeed, very high values of sorption capacity compared with the sorption of Cs and Sr were obtained for U. Also, this mechanism is possible during strontium sorption along with ion exchange, since the formation of low soluble strontium phosphate can occur under these conditions [53].

Some physicochemical data obtained for samples after sorption of U confirm the assumption about precipitation processes. (i) XRD patterns contain new peaks (they are marked with an asterisk in Figure 1), and they are more intense for the sample modified by HTT compared with the initial sample. These peaks are attributed to phase chernikovite (H-autunite) (H_3_O)_2_(UO_2_)_2_(PO_4_)_2_·6H_2_O (JCPDS 01-075-1106) [50,51,52,53,54,55]. At the same time, XRD patterns obtained for samples after Cs and Sr ions do not contain additional peaks. (ii) FTIR spectra recorded for these samples also indicate the formation of uranyl phosphate. This can be seen in Figure 6. Low-intensity bands at 994 and 948 cm^−1^ can be assigned to the symmetric stretching vibration of P-O and O=U=O bonds, respectively, in uranyl phosphate [51,56].

In addition, FTIR spectra in the range 2000–4000 cm^−1^, which are presented in Figure 2, show that the intensity and position of the band at 2400 cm^−1^ are changed. This band is assigned to vibrations of P-OH groups that are cation exchange sites. The observed shift of this band towards higher wavenumbers obviously indicates the ion-exchange sorption of cations. (iii) Finally, a decrease in the specific surface area and the total pore volume occurs as a result of sorption, as described in References [10,52]. For example, the decrease in the specific surface area after sorption of cesium, strontium, and uranium is 35, 59, and 171 m^2^/g, respectively, for samples modified by HTT of the wet gel at 250 °C (No. 10). It is important that this decrease is maximal for samples with sorbed uranium and minimal for samples with sorbed cesium. This fact indicates that the contribution of precipitation processes is maximal during the sorption of uranium.

## 4. Conclusions

Hydrothermal, microwave, and mechanochemical modifications allow us to prepare tin (IV) phosphate without micropores and with a developed meso–macroporous structure, practically without changing the crystal and surface structure. Therefore, the adsorption performance of modified samples in relation to Cs(I), Sr(II), and U(VI) cations depends critically on the parameters of the porous structure. At the same time, the mesoporous component, primarily the size of the mesopores, determines the sorption capacity of the modified samples in relation to the studied cations. It is this parameter that has the greatest effect on the size of the surface accessible for hydrated cations. The samples modified in the form of a wet gel showed the maximum sorption capacity, which confirms the effectiveness of this novel modification approach. In addition, sorption capacity, which increases in the row Cs, Sr, and U, depends on the mechanism of sorption. Cesium is sorbed exclusively by ion exchange. When strontium is sorbed, a precipitation mechanism is possible, although it is also mainly sorbed by the ion exchange way. On the other hand, precipitation processes play a predominant role in the sorption of uranium, as a result of which its sorption capacity is much higher than for the other two ions. The latter was confirmed by physicochemical studies of samples with sorbed uranium, namely by the formation of the uranyl phosphate phase in their structure. Obtained sorption results are well described by the Langmuir model, which is typical for other phosphates.

## Figures and Tables

**Figure 1 materials-16-04788-f001:**
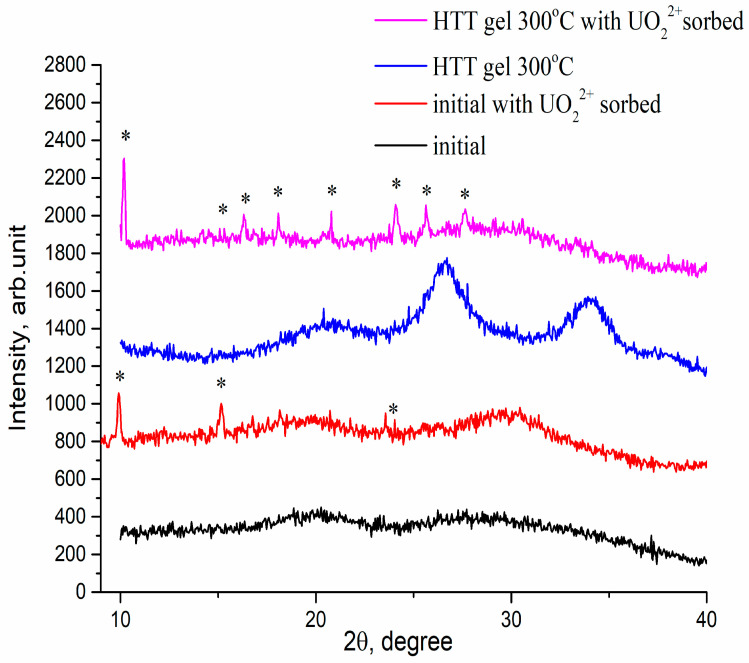
XRD patterns for some samples before and after sorption of uranyl ion.

**Figure 2 materials-16-04788-f002:**
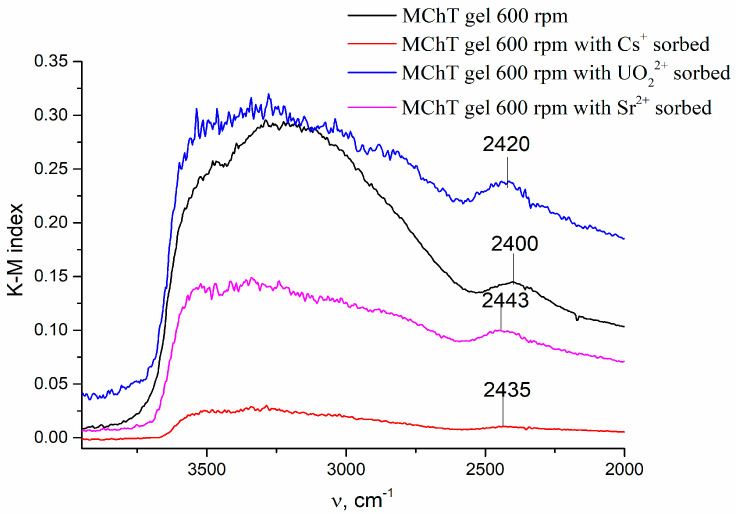
FTIR spectra for sample modified by MChT of wet gel at 600 rpm before and after sorption of cations.

**Figure 3 materials-16-04788-f003:**
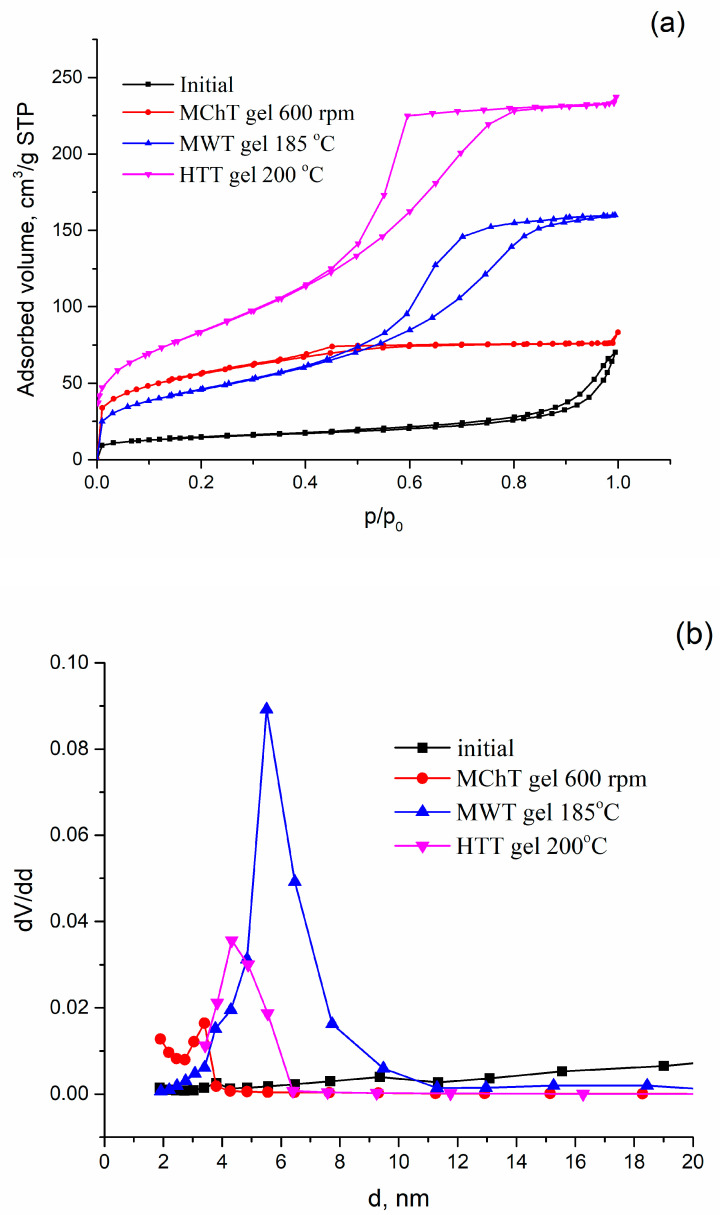
The nitrogen adsorption–desorption isotherms (**a**) and curves of pore size distribution (**b**) for some samples.

**Figure 4 materials-16-04788-f004:**
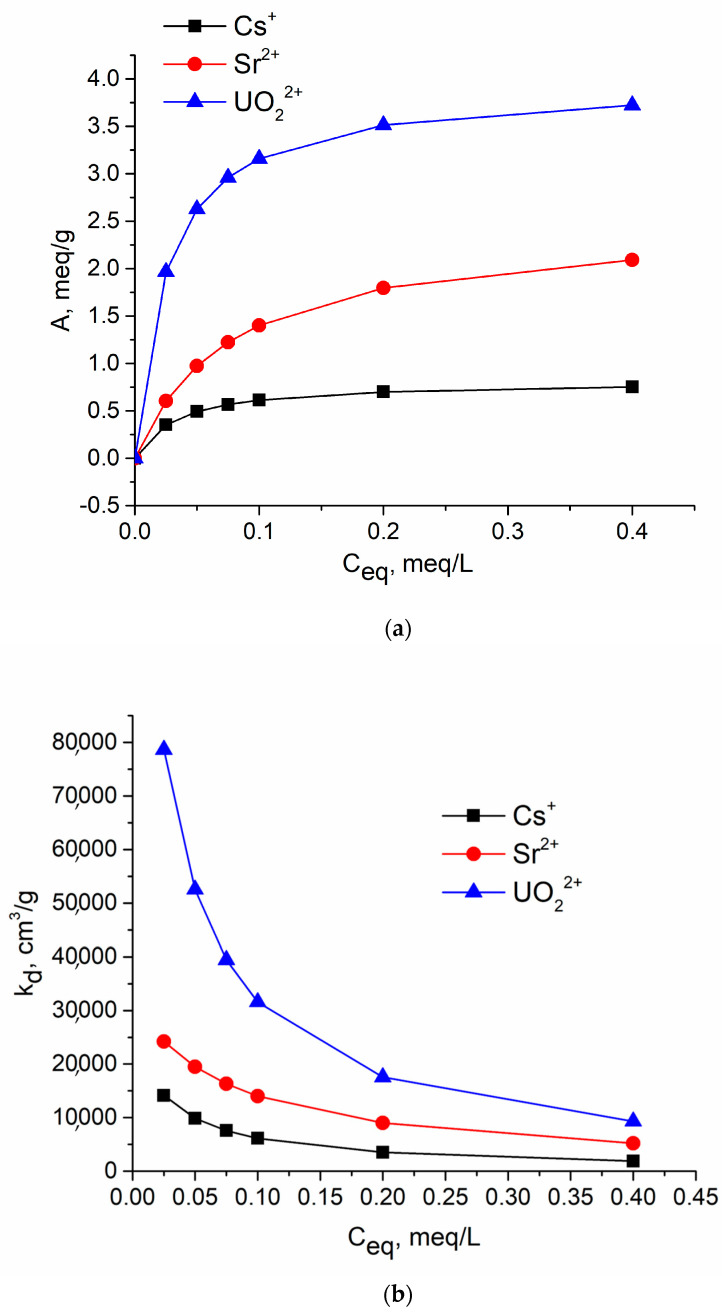
(**a**) The isotherms of sorption of Cs(I), Sr(II), and U(VI) on sample No. 8. (**b**) The dependences of distribution coefficient from equilibrium concentrations of Cs(I), Sr(II), and U(VI) for sample No. 8.

**Figure 5 materials-16-04788-f005:**
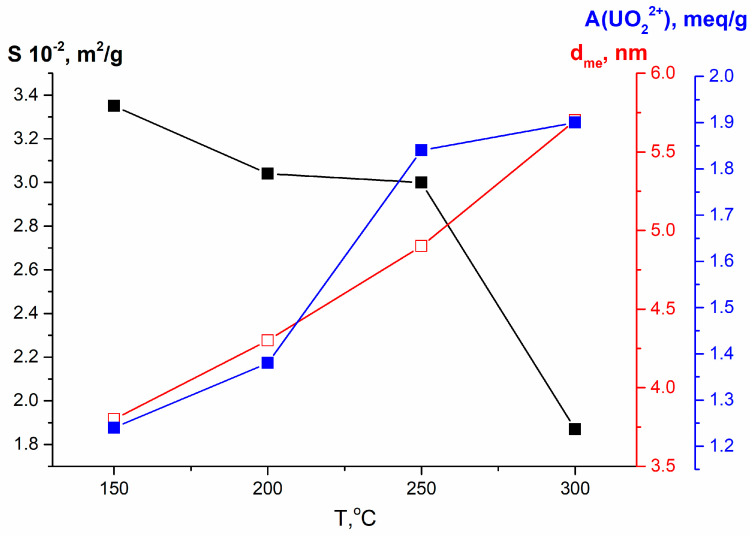
The dependencies of specific surface area, mesopore diameter, as well as sorption capacity in relation to U(VI) from temperature of HTT of wet gel.

**Figure 6 materials-16-04788-f006:**
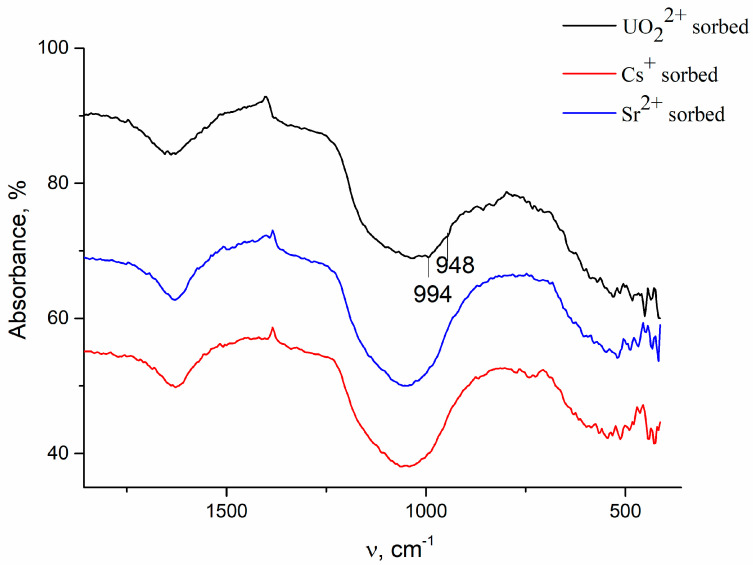
FTIR spectra for sample subjected to HTT of wet gel at 300 °C after sorption of cations.

**Table 1 materials-16-04788-t001:** The parameters of porous structure of studied samples.

N	Samples	S, m^2^/g	V_Σ_	V_me_	V_mi_	V_ma_	d, nm
			cm^3^/g				
1	Initial	50	0.11	0.10	0.01	-	3.0
2	MChT gel 600 rpm	208	0.23	0.05	0.08	0.10	3.3
3	MChT xerogel H_2_O 600 rpm	102	0.47	0.15	0.03	0.28	3.5; 40
4	MWT gel 185 °C	125	0.50	0.50	-	-	6.6
5	MWT xerogel 250 °C	62	0.31	0.16	-	0.15	4.3
6	HTT xerogel 200 °C	86	0.2	0.18	0.005	0.015	4.0
7	HTT xerogel 300 °C	19	0.32	0.075	-	0.245	7.7
8	HTT gel 150 °C	335	0.70	0.23	0.09	0.38	3.8
9	HTT gel 200 °C	304	0.53	0.34	0.02	0.16	4.3
10	HTT gel 250 °C	300	0.47	0.38	0.01	0.08	4.9
11	HTT gel 300 °C	187	0.46	0.27	-	0.18	5.7

**Table 2 materials-16-04788-t002:** Sorption characteristics for studied samples.

N	Samples	Cs		Sr		U	
		*A*, meq/g (% of Sorption)	Kd, cm^3^/g	*A*, meq/g (% of Sorption)	Kd, cm^3^/g	*A*, meq/g (% of Sorption)	Kd, cm^3^/g
1	Initial	0	0	0.052 (14.5)	323	0.44 (22.4)	586
2	MChT gel 600 rpm	0.20 (57.2)	2706	0.31 (80.7)	8477	1.64 (87.9)	13,838
3	MChT xerogel H_2_O 600 rpm	0.19 (55.5)	2446	0.28 (74.7)	5800	1.87 (98.9)	170,661
4	MWT gel 185 °C	0.12 (34.9)	1036	0.25 (67.5)	4004	1.67 (88.8)	15,261
5	MWT xerogel 250 °C	0	0	0.018 (4.8)	101	0.37 (19.1)	472
6	HTT xerogel 200 °C	0.094 (26.8)	783	0.17 (44.7)	1619	0.56 (29.5)	778
7	HTT xerogel 300 °C	0.049 (14.0)	350	0.080 (21.0)	533	0.26 (13.7)	299
8	HTT gel 150 °C	0.20 (60.7)	3011	0.31 (84.3)	10,496	1.24 (65.3)	3673
9	HTT gel 200 °C	0.21 (63.8)	3313	0.30 (83.1)	9282	1.38 (75.0)	5661
10	HTT gel 250 °C	0.20 (59.4)	2828	0.27 (74.7)	5711	1.84 (97.7)	83,043
11	HTT gel 300 °C	0.20 (59.8)	2847	0.32 (89.2)	15,721	1.90 (100)	—

**Table 3 materials-16-04788-t003:** The parameters of the Langmuir equation for sorption of Cs(I), Sr(II), and U(VI) on sample No. 8.

Ions	Q_0_, meq/g	b, L/meq	*A* = f(Ceq)	K_d_ = f(Ceq)
Cs	0.81	30.86	*A* = (1.23 + (0.040/C_eq_))^−1^	K_d_ = ((1.23 + (0.040/C_eq_))∙C_eq_)^−1^∙1000
Sr	2.50	12.90	*A* = (0.40 + (0.031/C_eq_))^−1^	K_d_ = ((0.40 + (0.031/C_eq_))∙C_eq_)^−1^∙1000
U	4.00	39.06	*A* = (0.25 + (0.0064/C_eq_))^−1^	K_d_ = ((0.25 + (0.0064/C_eq_))∙C_eq_)^−1^∙1000

## Data Availability

Not applicable.

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
