# Peer review of "The Effect of Hydrothermal, Microwave, and Mechanochemical Treatments of Tin Phosphate on Sorption of Some Cations"

_materials, 2023, doi:10.3390/ma16134788_

Round 1

Reviewer 1 Report

  Page 2, Line 79- Elsewhere is too general a comment. Please specify the name of auth

Line 82- Formula is incorrect. 

The work reported seems to be quite basic and may require more novelty.

 English language correction required. Major grammatical errors.

Author Response

The authors are grateful to the esteemed Reviewer for the useful remarks and comments to the manuscript. We took into account the comments and suggestions of the Reviewer as much as possible, based on the deadline offered to us for correcting the manuscript. All changes and additions to the text are highlighted in red.

Comments and Suggestions for Authors

  • Page 2, Line 79- Elsewhere is too general a comment. Please specify the name of auth

Unfortunately, we did not understand this remark.

  • Line 82- Formula is incorrect. 

The decipherment of this abbreviation is given on p. 1 (line 31).

  • The work reported seems to be quite basic and may require more novelty.

Indeed, at first glance, the statement about the influence of the porous structure on the sorption properties of tin phosphate is trivial. However, there are not many studies illustrating this effect for specific sorbents in specific ion exchange processes. This applies even more to such a microporous material as precipitated tin phosphate.

The proposed modification methods make it possible to form a mesoporous or meso-macroporous structure that is more accessible to cations. In this regard, modification of wet gel before its drying opens up particularly wide possibilities. This is a new approach for modifying precipitated tin phosphate is noted in the Introduction (p.2), as well as in Abstract  and Conclusions.

Reviewer 2 Report

This work by Khalameida and co-workers describe the effect of different means of pretreatment of Sn(IV) phosphate and their adsorption performance towards metal cations like Cs(I), Sr(II) and U(VI). The main effect is on porosity, and then the authors elucidated the mechanism of adsorption mainly by X-ray diffraction. On balave my assessment is positive, and I recommend minor revision of the following work according to the following points.

1) Authors should explicitly write the oxidation state of tin in the sorbents, in the abstract and the text.

2) Figure 5. I am not convinced by the discussion of the FTIR spectra. There is actually no band at 948 cm-1, the spectra are too low in resolution to make this assignment. Please obtain a better FTIR spectrum for the sorbent with uranyl cation, otherwise remove the comment.

Author Response

The authors are grateful to the esteemed Reviewer for the useful remarks and comments to the manuscript. We took into account the comments and suggestions of the Reviewer as much as possible, based on the deadline offered to us for correcting the manuscript. All changes and additions to the text are highlighted in red.

Comments and Suggestions for Authors

This work by Khalameida and co-workers describe the effect of different means of pretreatment of Sn(IV) phosphate and their adsorption performance towards metal cations like Cs(I), Sr(II) and U(VI). The main effect is on porosity, and then the authors elucidated the mechanism of adsorption mainly by X-ray diffraction. On balave my assessment is positive, and I recommend minor revision of the following work according to the following points.

  • Authors should explicitly write the oxidation state of tin in the sorbents, in the abstract and the text.

This clarification has been made.

2) Figure 5. I am not convinced by the discussion of the FTIR spectra. There is actually no band at 948 cm-1, the spectra are too low in resolution to make this assignment. Please obtain a better FTIR spectrum for the sorbent with uranyl cation, otherwise remove the comment.

In our opinion, we can talk about the presence of this strip as a shoulder. Moreover, this result coincides with literature data [51,56]. Unfortunately, we did not have enough time to repeat these measurements (we had 10 days to respond to the reviewers' comments and make changes to the manuscript).

Reviewer 3 Report

Authors can add more details about microwave treatment such as duration and the amount of sample.

Temperature plays a crucial role in sample treatment. Did the authors measure the temperature during microwave treatment?

Author Response

The authors are grateful to the esteemed Reviewer for the useful remarks and comments to the manuscript. We took into account the comments and suggestions of the Reviewer as much as possible, based on the deadline offered to us for correcting the manuscript. All changes and additions to the text are highlighted in red.

Comments and Suggestions for Authors

  • Authors can add more details about microwave treatment such as duration and the amount of sample.

Temperature and duration of microwave modification were 185°C and 0.5 h in the case of wet gel treatment and 250°C and 0.5 h in the case of dried xerogel treatment. This is indicated on p. 2.

  • Temperature plays a crucial role in sample treatment. Did the authors measure the temperature during microwave treatment?

“NANO 2000” high–pressure reactor, used for microwave treatment, is equipped with temperature and pressure sensors and continuous recording of their changes during processing.

Reviewer 4 Report

In this study, the researchers investigated the adsorption behavior of Cs, Sr, and U on tin phosphate (SnP) and employed various characterization techniques such as XRD, XRF, FTIR, and adsorption measurements. They also applied three different modification techniques, namely mechanochemical treatment (MCT), hydrothermal treatment (HTT), and microwave treatment (MWT). The article presented a comprehensive scientific analysis; however, a few points need to be addressed before considering it for publication:

-          The scope of investigated adsorption types seems limited. It would be beneficial if the authors discussed the potential forms of interaction between these metals and SnP, possibly through the use of illustrative figures.

-          It would be valuable for the authors to describe the adsorption of Cs, Sr, and U on SnP using the Langmuir and Elovich methods, as previously demonstrated in a study on titanium phosphate by author (ref 24).

-          Regarding the MTT method, the authors should provide a rationale for choosing a rotation speed of 600 rpm for the experimentation.

-          The authors should clarify the reason behind studying the adsorption (MWT) at 185°C and also consider including information on the adsorption at 300°C.

Addressing these points will strengthen the manuscript and improve its contribution to the field.

Minor editing of English language required

Author Response

The authors are grateful to the esteemed Reviewer for the useful remarks and comments to the manuscript. We took into account the comments and suggestions of the Reviewer as much as possible, based on the deadline offered to us for correcting the manuscript. All changes and additions to the text are highlighted in red.

Comments and Suggestions for Authors

In this study, the researchers investigated the adsorption behavior of Cs, Sr, and U on tin phosphate (SnP) and employed various characterization techniques such as XRD, XRF, FTIR, and adsorption measurements. They also applied three different modification techniques, namely mechanochemical treatment (MCT), hydrothermal treatment (HTT), and microwave treatment (MWT). The article presented a comprehensive scientific analysis; however, a few points need to be addressed before considering it for publication:

-          The scope of investigated adsorption types seems limited. It would be beneficial if the authors discussed the potential forms of interaction between these metals and SnP, possibly through the use of illustrative figures.

The text of the manuscript contains a certain discussion of the mechanisms of sorption of cations on the surface of tin phosphate (p. 9). These are ion-exchange sorption and precipitation of insoluble compounds of cations. In both cases, interaction of sorbed cation with P-OH groups takes place. Thus, cesium ions are sorbed exclusively by ion exchange through the substitution of protons in acidic hydroxyl groups.  On the contrary, uranyl ions are deposited on the surface in the form of insoluble phosphates. The results of FTIR and XRD studies for spent sorbents support these possible mechanisms of cation binding on the tin phosphate surface (p. 10-11).

-          It would be valuable for the authors to describe the adsorption of Cs, Sr, and U on SnP using the Langmuir and Elovich methods, as previously demonstrated in a study on titanium phosphate by author (ref 24).

The isotherms of sorption for all tested ions were obtained on sample 8 which have the most developed porous structure. These isotherms were processed using Freundlich, Langmuir and Dubinin-Radushkevich models and shown that experimental data are best described by the Langmuir equation. Figures 4a, 4b, Table 3 and discussion of the results are added to the text of the manuscript (p. 7-9).

-          Regarding the MTT method, the authors should provide a rationale for choosing a rotation speed of 600 rpm for the experimentation.

   As noted on p. 3, samples with a high specific surface area and a meso-macroporous structure (p. 3), obtained by various methods described in paper [18], were selected for sorption measurements. Among the samples modified by the mechanochemical method, the sample milled at 600 rpm in the form of a gel meets these requirements.

Corresponding additions are made on p. 3.

-          The authors should clarify the reason behind studying the adsorption (MWT) at 185°C and also consider including information on the adsorption at 300°C.

“NANO 2000” high–pressure reactor, used for microwave treatment, can be used at a temperature not higher than 270°C. Therefore, the modification temperature is impossible for this type of reactor.

 Addressing these points will strengthen the manuscript and improve its contribution to the field.
